

# Aerosol optical depth trend over the Middle East

**K. Klingmüller**[1], **A. Pozzer**[1], **S. Metzger**[2], **G. Stenchikov**[3], **and J. Lelieveld**[1,2]

[1]Max Planck Institute for Chemistry, P.O. Box 3060, 55020 Mainz, Germany
[2]The Cyprus Institute, P.O. Box 27456, 1645 Nicosia, Cyprus
[3]King Abdullah University of Science and Technology, Thuwal 23955-6900, Saudi Arabia

Correspondence to: K. Klingmüller (k.klingmueller@mpic.de)

Received: 16 October 2015 – Accepted: 9 December 2015 – Published: 15 January 2016

Published by Copernicus Publications on behalf of the European Geosciences Union.

ACPD

doi:10.5194/acp-2015-839

AOD trend over the Middle East

K. Klingmüller et al.

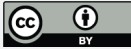

Discussion Paper | Discussion Paper | Discussion Paper | Discussion Paper

**ACPD**

doi:10.5194/acp-2015-839

**AOD trend over the Middle East**

K. Klingmüller et al.

## Abstract

We use the combined Dark Target/Deep Blue aerosol optical depth (AOD) satellite product of the Moderate-resolution Imaging Spectroradiometer (MODIS) collection 6 to study trends over the Middle East between 2000 and 2015. Our analysis corroborates a previously identified positive AOD trend over large parts of the Middle East during the period 2001 to 2012.

We relate the annual AOD to precipitation, soil moisture and surface winds to identify regions where these attributes are directly related to the AOD over Saudi Arabia, Iraq and Iran. Regarding precipitation and soil moisture, a relatively small area in and surrounding Iraq turns out to be of prime importance for the AOD over these countries. Regarding surface wind speed, the African Red Sea coastal area is relevant for the Saudi Arabian AOD.

Using multiple linear regression we show that AOD trends and interannual variability can be attributed to soil moisture, precipitation and surface winds, being the main factors controlling the dust cycle. Our results confirm the dust driven AOD trends and variability, supported by a decreasing MODIS-derived Ångström exponent and a decreasing AERONET-derived fine mode fraction that accompany the AOD increase over Saudi Arabia. The positive AOD trend relates to a negative soil moisture trend. As a lower soil moisture translates into enhanced dust emissions, it is not needed to assume growing anthropogenic aerosol and aerosol precursor emissions to explain the observations. Instead, our results suggest that increasing temperature and decreasing relative humidity in the last decade have promoted soil drying, leading to increased dust emissions and AOD; consequently an AOD increase is expected due to climate change.

Discussion Paper | Discussion Paper | Discussion Paper | Discussion Paper

## 1 Introduction

The Middle East and the adjacent Mediterranean region have been identified as a hot-spot of climate change (Hoerling et al., 2012; Hemming et al., 2010; Lelieveld et al., 2012; Giorgi and Lionello, 2008). These studies indicate a relatively strong precipitation decrease and temperature increase in the near future, which would be especially important for an arid and desert dominated region such as the Middle East. Since the middle of the 20th century, the annual number of unusually hot days and nights in the Middle East has already increased whereas the number of cool days and nights has decreased significantly (IPCC, 2014).

Beside the climatic importance of the region, the Middle East is also outstanding from an atmospheric chemistry point of view. High levels of ozone are expected due to the strong insolation acting on local pollution emissions, in addition to long range transport and stratospheric ozone intrusions (Lelieveld et al., 2009; Zanis et al., 2014). Model projections of future anthropogenic emissions suggest that this situation will likely exacerbate in the near future (Pozzer et al., 2012) although in the past few years pollution emissions have decreased in the aftermath of economic crises and conflicts in the region (Lelieveld et al., 2015). Nevertheless, pollution levels are very high, including aerosols that add to a high background of natural particles, leading to more than 100 000 premature deaths per year in the Middle East (Giannadaki et al., 2014).

The Middle East is centrally located in the so called dust-belt (Astitha et al., 2012) and strongly influenced by natural sources. The high atmospheric dust concentrations near the Earth's surface are reflected in high aerosol optical depth (AOD) levels (e.g., Hsu et al., 2012). Increasingly, the natural dust is supplemented by local anthropogenic emissions associated with the economic and population growth on the Arabian Peninsula which adds to pollution originating from Europe, Asia and Africa. Polluted dust can significantly enhance atmospheric heating (Mishra et al., 2014), especially by intense incoming solar radiation that is typical in the Middle East. On the other hand, chemical ageing in polluted dust air masses transported over long distances can in-

Discussion Paper | Discussion Paper | Discussion Paper | Discussion Paper |

ACPD

doi:10.5194/acp-2015-839

**AOD trend over the Middle East**

K. Klingmüller et al.

**ACPD**

doi:10.5194/acp-2015-839

crease deposition rates of dust particles (Abdelkader et al., 2015). Various studies show a strong increase in the AOD over the Middle East during the last decade. These studies were performed with numerical models (Pozzer et al., 2015) and with different remotely sensed observational data, e.g., from SeaWiFS (Hsu et al., 2012; IPCC, 2014), MODIS, MISR and AERONET (de Meij et al., 2012; de Meij and Lelieveld, 2011; Stevens and Schwartz, 2012).

Different from previous studies, in this work we analyse trends using the most recent (as of 2015) data collection 6 of the Moderate-resolution Imaging Spectroradiometer (MODIS) (Hubanks et al., 2015; Levy et al., 2013; MODIS MOD08 M3, 2015) which includes AOD data based on refined retrieval algorithms (Sayer et al., 2014), in particular the expanded Deep Blue algorithm. The latter is of special importance for our study because it is suited for retrievals over bright surfaces such as the deserts and semi-deserts covering large parts of the Middle East. Among other improvements, the new data version provides extended spatiotemporal coverage. In addition, it introduces a merged AOD product, combining retrievals from the Deep Blue and Dark Target algorithms to produce a consistent data set covering a multitude of surface types ranging from oceans to bright deserts. A crucial aspect is the problem of spurious AOD trends due to instrumental drift found in earlier MODIS collections (Wang et al., 2012; Zhang and Reid, 2010; Levy et al., 2010), which has been addressed recently (Levy et al., 2013). Another difference to previous studies is the extended time period considered: to verify the persistence of the AOD increase, we take MODIS data up to 2015 into account.

A second objective of this study is to shed light on the causes of the observed trends. In view of changing anthropogenic aerosol emissions, a crucial question is whether these are responsible for the AOD trends or if they are related to natural dust. The observed aerosol properties and the number of dust events seem to favour the latter. Based on station data, Notaro et al. (2015) even define a regime shift around 2006 from an inactive to an active dust period which they attribute to synergistic interactions between the El Niño Southern Oscillation (ENSO) and the Pacific Decadal Oscillation

(PDO), superimposed onto a long-term drying trend in the Fertile Crescent. To explore further the link between AOD trend and increased dust activity, we relate the MODIS AOD to observations of major parameters controlling the dust cycle: precipitation, surface wind and soil moisture.

This article is structured as follows: In Sect. 2 the observed aerosol trends are presented and discussed. The AOD is related to precipitation, soil moisture and surface wind speed individually in Sects. 3, 4 and 5. In addition, trends of these variables are discussed. Finally, in Sect. 6 a multivariable linear model based on all three observables is used to reproduce the MODIS AOD trends, leading to our conclusions in Sect. 7.

## 2  Aerosol trends

Figure 1 presents a global overview of AOD trends over 15 years between March 2000 and February 2015. It has been computed from the monthly values of the combined Dark Target/Deep Blue 550 nm AOD from MODIS Terra, collection 6 (Hubanks et al., 2015; MODIS MOD08 M3, 2015). For deseasonalisation, the annual cycle obtained by harmonic regression of sixth order has been subtracted. The trend of the deseasonalised AOD ist calculated by fitting a linear model using generalised least squares taking into account the time series auto correlation with a time lag of one month (Pinheiro et al., 2015).

Strong and significant positive trends extending over large areas are found in the Middle East, in particular the Arabian Peninsula. Comparably strong trends are only found in China and, though restricted to a small area, under the exceptional conditions in the vicinity of the Aral Sea. While the ten year trend from January 2001 to December 2010 considered by Pozzer et al. (2015) (see also Fig. S1 in the Supplement) clearly identifies the Middle East as having the strongest AOD increase worldwide, it is surpassed by the corresponding trend over China when the period up to 2015 is also included.

Discussion Paper | Discussion Paper | Discussion Paper | Discussion Paper | Discussion Paper |

**[ACPD](doi:10.5194/acp-2015-839)**

doi:10.5194/acp-2015-839

**AOD trend over the Middle East**

K. Klingmüller et al.

Nevertheless, the Middle East is clearly a hot spot of AOD increase. Magnifying this region, the left panel of Fig. 2 reveals large areas with an average increase in excess of $0.01\,\text{year}^{-1}$, and regional increase rates higher than $0.02\,\text{year}^{-1}$. Over major parts of Saudi Arabia the positive AOD trend is significant with probability ($p$) values of the

observations (using the trendless case as null hypothesis) below the significance level of 1 %. Compared to the ten year trend from 2001 to 2010 (Fig. S2 in the Supplement), in particular over Iraq, northern Saudi Arabia and the Persian Gulf the AOD increase is less significant with $p$ values exceeding 1 %.

In the following we mostly consider AOD values spatially averaged over the territory

of selected countries. The advantage is that boundaries of countries are well defined, whereas regions of strong AOD trends, for example, might vary over time. Country level trends might also be most relevant for policy makers, e.g., in view of air quality standards.

Figure 3 shows the evolution of the MODIS AOD averaged over Saudi Arabia. The

figure reveals that the AOD increase is limited to the period between 2001 and 2012. The AOD levels in the year 2000 are high compared to the subsequent years, and the levels in 2013 and especially in 2014 are low compared to the preceding years. Systematically varying length and centre of the time interval used for the trend analysis (see Fig. S4 in the Supplement) further supports that the most significant AOD

increase is observed for time intervals centred around the year 2007, which coincides with the regime shift from an inactive to an active dust period around 2006 defined by Notaro et al. (2015). The same applies to the MODIS AOD trends over Iraq and Iran (Figs. S6 and S8 in the Supplement), so that in the following our trend analysis primarily considers the twelve year period between 2001 and 2012. Note that in particular

the end of the period is not distinctly defined. For example, over Iraq a strong increase until 2008 is observed, whereas from 2008 to 2012 the deseasonalised AOD remains approximately constant on a high level (Fig. S5 in the Supplement). The right panel of Fig. 2 displays the Middle Eastern pattern of the twelve year trends, which is very similar to the ten year pattern shown in Fig. S1.

**ACPD**

doi:10.5194/acp-2015-839

**AOD trend over the Middle East**

K. Klingmüller et al.

Discussion Paper | Discussion Paper | Discussion Paper | Discussion Paper |

For the 2001 to 2012 period, our trend analysis yields an average annual AOD increase of $0.0135 \pm 0.0020$ over Saudi Arabia (see Fig. 3). Consistent with Hsu et al. (2012), the winter AOD levels are relatively invariant; exceptions are the very low levels during the winters 2000/2001 and 2001/2002. This hints at dust being the aerosol component mainly responsible for the perennial AOD variability.

There are numerous AERONET stations in the Middle East (Holben et al., 1998; AERONET, 2015), however only few have data records extending over ten years or more (Fig. S9 in the Supplement). Only one, Solar Village, is located in a region of, according to the MODIS data, strong and significant positive AOD trend.

From April 2000 to April 2013 the Solar Village data overlaps with MODIS data. The AERONET AOD values, shown in Fig. 4, exhibit a similar trend as the MODIS AOD in Saudi Arabia (Fig. 3). The fluctuations of the two time series share common features and both, the annual AOD increase and the corresponding $p$ value, are of comparable magnitude. Interestingly, the two closest AERONET stations with long-term data records, the Israeli stations at Nes Ziona and Sde Boker do not observe a significant AOD trend, see Figs. S11–S14 in the Supplement.

The increase of AOD in Saudi Arabia during the last decade is accompanied by a decrease of the Ångström exponent, which is inversely related to the size of the particles. The MODIS Ångström exponent shown in Fig. 5 is noticeably anti-correlated to the AOD: during the dusty high AOD periods in spring and summer, the Ångström exponent reaches annual minima. Moreover, the maxima in the deseasonalised AOD of Fig. 3 (2001, 2004, 2008 to 2009, 2011 to 2012) correspond to minima in the deseasonalised Ångström exponent. The correlation coefficient of the deseasonalised monthly values between March 2000 and February 2015 is −0.79. The decreasing Ångström exponent clearly indicates a trend from 2001 to 2012 towards relatively large aerosol particles, suggesting an increased amount of coarse mode dust aerosols and generally a dominant role of dust regarding the AOD variability. The same conclusion can be drawn from the AERONET fine mode fraction shown in Fig. 6 which exhibits

Discussion Paper | Discussion Paper | Discussion Paper | Discussion Paper |

**ACPD**

doi:10.5194/acp-2015-839

**AOD trend over the Middle East**

K. Klingmüller et al.

a significant negative trend (in contrast to the fine mode fractions at the Israeli stations, Figs. S16–S19 in the Supplement).

To further investigate the potentially dominant role of dust and possible reasons for the AOD increase, in the following we study major parameters that control the dust cycle: precipitation, surface wind and soil conditions, the latter represented by the surface soil moisture. In this study, we rely solely on correlation analysis. Deeper insight into the causality between the observables as well as the role of atmospheric transport will be provided by future modelling studies.

## 3  Precipitation

Precipitation can affect the aerosol and especially the dust load via several mechanisms: precipitation scavenging can remove aerosol particles from the atmosphere, increased soil moisture reduces wind induced dust emissions and additionally fosters vegetation growth, which further inhibits dust emissions. Hence, a trend in the precipitation rate could help explain the AOD increase in the Middle East.

We use the TRMM 3B42 precipitation data (Huffman et al., 2007; TRMM 3B42, 2015) to study the relation between precipitation and AOD. Annual average values of the precipitation rate are used, where the averaging period starts with the precipitation season of the region on 1 September, and the precipitation data is re-gridded to a coarser 2° grid. The annual average AOD values are calculated with an averaging period starting with the regional AOD season on 1 December. The time shift between the averaging periods used for precipitation and AOD allows for effects of vegetation growth and soil moistening during autumn. For Saudi Arabia, Iraq and Iran the correlation of the individual precipitation pixels with the AOD, spatially averaged over the country territories, is computed. The resulting three correlation maps are shown in the top row of Fig. 7. For all three countries the AOD is significantly anticorrelated to the precipitation in Iraq and northern Saudi Arabia with absolute values of the correlation coefficients regionally exceeding 0.8.

Discussion Paper | Discussion Paper | Discussion Paper | Discussion Paper |

**ACPD**

doi:10.5194/acp-2015-839

**AOD trend over the Middle East**

K. Klingmüller et al.

For the precipitation trend analysis, in addition to the TRMM 3B42 data, we consider TRMM 3B31 data (Huffman et al., 2007; TRMM 3B31, 2015), the three CMORPH (Joyce et al., 2004; CMORPH V1.0, 2015) variants RAW, ADJ, BLD (beta) and the enhanced version of the CMAP data (Xie and Arkin, 1997; CMAP, 2015). We use the same analysis as for the MODIS AOD (Sect. 2). For each data set, Fig. S22 in the Supplement shows the resulting trend pattern over the Middle East. None individually shows a significant trend, neither positive nor negative, extending over a larger region. Regionally, $p$ values below 0.01 are found, but the affected regions are not consistent for the different data sets and it is evident that the precipitation does not exhibit the same distinct trend as the AOD, cf. Fig. 2. This is confirmed by Fig. S23 in the Supplement, where annual averaged instead of deseasonalised precipitation values are considered which is expected to yield more robust results in regions of sporadic precipitation. We conclude that even though the AOD in the Middle East can be strongly linked to regional precipitation, changes in precipitation alone are unlikely to be the reason for the positive AOD trend.

## 4   Soil moisture

A long term global surface soil moisture (SSM) data record based on satellite mounted active and passive microwave sensors is provided by the European Space Agency Climate Initiative (ESA-CCI) (Liu et al., 2012, 2011; Wagner et al., 2012; ESA-CCI soil moisture, 2015). We use the COMBINED data set which covers a time period up to and including the year 2013.

We perform an AOD–soil-moisture correlation analysis analogously to the AOD–precipitation analysis above. For both, AOD and soil moisture, we use annual average values where the averaging period starts with the AOD season on 1 December. The soil-moisture data is re-gridded to the coarser 2° grid. For Saudi Arabia, Iraq and Iran the correlation of the individual soil-moisture pixels with the AOD, spatially averaged over the national territory, is computed. The resulting three correlation maps are

Discussion Paper | Discussion Paper | Discussion Paper | Discussion Paper |

**ACPD**

doi:10.5194/acp-2015-839

**AOD trend over the Middle East**

K. Klingmüller et al.

shown in the bottom row of Fig. 7. In all three cases we find strong anti-correlations of the AOD to the soil moisture in Iraq and surrounding areas, with absolute values of the correlation coefficients above 0.8.

Again we use the same analysis as for the MODIS AOD (Sect. 2) to analyse trends in the soil moisture data. Figure S24 in the Supplement shows significant negative trends in large areas of Syria, Iraq and Saudi Arabia. As the AOD is anti-correlated with soil moisture, this could translate into the observed positive AOD trend. Since we do not observe an accompanying trend with comparable significance in the precipitation data, we assume that the soil moisture trend is predominantly caused by increased evaporation due to increasing temperatures (Fig. S26 in the Supplement) and a resulting decrease of the relative humidity (Fig. S27 in the Supplement), but possibly also due to changing land use, and that soil moisture reflects the increasing drought conditions in the Fertile Crescent during the last decade more distinctively than the precipitation rate.

## 5 Surface wind

Another important factor impacting dust emissions is the surface wind which drives the saltation bombardment mechanism (Shao et al., 1993). To study this we use the ERA-Interim data for wind speed at 10 m altitude (Dee et al., 2011) and perform the same analysis as for the soil moisture. Figure 8 shows the correlation of the AOD over Saudi Arabia and Iraq with the wind speed over coastal regions of the Red Sea, and the AOD over Iran with the wind speed over coastal regions of the Persian Gulf. The regions with significant correlation are smaller than the corresponding regions for soil moisture and precipitation. However, the region around the Hala'ib Triangle on the African Red Sea coast, where the wind speed is correlated to the AOD over Saudi Arabia and the Iraq, coincides with a region of positive surface wind speed trend (see Fig. S28 in the Supplement) so that it contributes to the observed AOD increase, e.g., over the Red Sea. In contrast, the surface wind speed in the Fertile Crescent did not increase

**[ACPD](doi:10.5194/acp-2015-839)**

doi:10.5194/acp-2015-839

**AOD trend over the Middle East**

K. Klingmüller et al.

and regionally even decreased. Therefore, in agreement with Notaro et al. (2015), we conclude that it does not notable contribute to increased dust activity.

## 6 Multiple linear regression analysis

From the previous sections we conclude that the AOD is correlated to all three variables considered, precipitation, soil moisture and surface wind. Especially soil moisture exhibits a significant trend which could explain the AOD increase, possibly supported by the regional positive surface wind trend. For further analysis regarding the impact of each of these factors, we employ multivariable linear models for the annual AOD.

The annual values averaged over the yellow marked pixels in Figs. 7 and 8, for which the correlation indices are below −0.8 or above 0.8, respectively, are used as predictors representing precipitation, soil moisture and surface wind; the response variable is the annual AOD over Saudi Arabia, Iraq and Iran. The same averaging periods as above are used (for precipitation starting on 1 September, for the other variables starting on 1 December). For each country, the most general model we consider takes the form

$$\tau = \beta_0 + \beta_\theta \theta + \beta_P P + \beta_\mathrm{w} w, \tag{1}$$

where $\tau$ is the AOD, $\theta$ the volumetric water content of the surface soil, $P$ the precipitation rate and $w$ the surface wind speed. The regression coefficients $\beta_0$, $\beta_\theta$, $\beta_P$ and $\beta_\mathrm{w}$ are determined by the least squares method. A similar model has been used by Yu et al. (2015) for seasonal dust prediction based on precipitation rates and sea surface temperatures.

In Table 1 we compare four variants of the model. The corresponding scatter plots are shown in Fig. S30 in the Supplement. We initially focus on the inhibiting factors precipitation and soil moisture and omit the wind speed term $\beta_\mathrm{w} w$ in Eq. (1). The resulting model and the two possible submodels using only one predictor, either soil moisture or precipitation, are represented by the first three rows of the table. Overall, the very good agreement between the modelled values and the observations, reflected by high

**ACPD**

doi:10.5194/acp-2015-839

**AOD trend over the Middle East**

K. Klingmüller et al.

$R^2$ values, is striking. For all three countries, the model using both predictors yields the highest $R^2$ values. For Saudi Arabia however, using the soil moisture alone yields a comparable $R^2$ value. The small-sample-size corrected Akaike information criterion (AICc) value, which includes a penalty for additional estimated parameters, suggests that in Saudi Arabia soil moisture is the dominant factor and even that precipitation could be omitted from the linear model. Also for Iraq soil moisture is a more relevant predictor than precipitation, but taking into account that precipitation improves the model significantly using both predictors is justified. For Iran we find the reverse relationship, where precipitation is the more relevant predictor, but also here using both variables yields the optimal model. The importance of precipitation for the Iranian AOD indicates a more important role of aerosol transport and precipitation scavenging along the way.

The relevance of each predictor for the different countries becomes even more apparent in Fig. 9. It compares the AOD observed by MODIS with the values resulting from the linear model using soil moisture and precipitation as predictors (third row in Table 1), breaking down the variability contributions of the soil moisture term $\beta_\theta\theta$ and the precipitation term $\beta_P P$. The corresponding regression coefficients are presented in Table 2. Again, for all three countries, the agreement of model and observation is apparent, suggesting that the interannual variability as well as the observed AOD trends, in particular the increase from 2001 to 2009, can be attributed to soil properties and precipitation that control the dust emissions and removal. Moreover, the individual contributions of the soil moisture term $\beta_\theta\theta$ and the precipitation term $\beta_P P$ indicate that the decreasing soil moisture from 2001 to 2009 is the driving force behind the observed AOD increase during the same period, especially in Saudi Arabia and Iraq. For Iran, the precipitation term contributes more to the AOD variability than soil moisture which, as mentioned above, could be explained by a more important role of aerosol transport and associated precipitation scavenging.

Even though the model using soil moisture and precipitation as predictors performs well for Saudi Arabia with an $R^2$ value of 0.85, it performs less good than for Iraq and

**ACPD**

doi:10.5194/acp-2015-839

**AOD trend over the Middle East**

K. Klingmüller et al.

Iran for which it yields $R^2$ values of 0.94. A considerable improvement can be achieved by taking into account the wind speed as a third predictor. As displayed in the last row of Table 1 the $R^2$ value is enhanced to 0.93 and the AICc value is reduced. This corroborates the impact of dust emissions from the African Red Sea coast on the Saudi

Arabian AOD (Kalenderski et al., 2013). It has been reported that dust generated and channelled through the Tokar Gap on the Red Sea coast of Sudan is frequently crossing the Red Sea and reaches Saudi Arabia (Jiang et al., 2009). The region identified by our method is located slightly further north along the coast. The relevance of this region for the AOD averaged over the whole Saudi Arabian territory might be related

to its location north of the highest peaks of the Asir Mountains which are a natural barrier for atmospheric transport, extending along the south-western coast of Saudi Arabia. An analysis of wind patterns at 700 hPa altitude that corroborates the importance of westerly winds for Saudi Arabia, in particular during winter, is shown in Fig. S29 in the Supplement. A more detailed analysis of the atmospheric transport involved

and the response of the African dust emissions to the surface wind trends remains to be conducted in a future modelling study using the atmospheric chemistry-climate model EMAC as used by Abdelkader et al. (2015) which includes online dust emission schemes (Tegen, 2002; Balkanski et al., 2004; Astitha et al., 2012) and the option to assimilate meteorological parameters, e.g., from the ERA-Interim re-analyses. Unlike

for Saudi Arabia, for Iraq and Iran including the surface wind as predictor does not improve the model.

## 7  Conclusions

Using the combined Dark Target/Deep Blue AOD products of the recently available MODIS collection 6, our study corroborates the positive AOD trend over the Middle East

during the 10 year period from 2001 to 2010, which has been reported previously based on other satellite products. Extending the trend analysis to a time period of 15 years from 2000 to 2015 shows that this trend persists until approximately the year 2012.

**ACPD**

doi:10.5194/acp-2015-839

**AOD trend over the Middle East**

K. Klingmüller et al.

Since then it is interrupted by lower AOD values in subsequent years. Considering that the AOD increase over Saudi Arabia is accompanied by a decreasing MODIS Ångström exponent and a decreasing AERONET fine mode fraction at the Solar Village station, coarse mode dust particles are likely the main contributors to the positive AOD trend.

We have identified regions where precipitation, soil moisture and surface winds are correlated with the AOD over Saudi Arabia, Iraq and Iran. Regarding precipitation and soil moisture, the relevant regions are located in Iraq and adjacent areas, regardless over which country the AOD is considered, identifying this central region, largely coincident with the Fertile Crescent, as crucial for the Middle Eastern AOD. Owing to transport by northwesterly Shamal winds its dust activity affects the whole Arabian Peninsula (Notaro et al., 2013, 2015).

Our multivariable regression analysis shows that the AOD trend and interannual variability can be related to variations of soil moisture, precipitation and surface winds in the particular regions. The positive AOD trend over Saudi Arabia and Iraq relates to a negative soil moisture trend. The AOD over Iran appears to be more strongly affected by precipitation, which points to a more important role by aerosol transported into the country crossing areas with significant precipitation, e.g., over the Zagros Mountains. For Saudi Arabia the increasing surface wind near the African coast of the Red Sea seems to be additionally involved.

We conclude that our results confirm a dust driven AOD trend and variability, and that additional anthropogenic aerosol emissions are likely less relevant for the AOD increase observed in the last decade. However, as the increased dust emissions are likely related to drought conditions due to anomalously high temperatures, a contribution by climate change is expected. It seems that dust emissions sensitively respond to increasing temperature, which reduces the relative humidity, thus enhancing evaporation and promoting soil drying. We note that changing industrial emissions can alter aerosol properties which are not considered in this study. For example, mixing with black carbon decreases the single scattering albedo without notably affecting the AOD (Klingmüller et al., 2014) and the interaction of dust and air pollution can enhance hy-

**ACPD**

doi:10.5194/acp-2015-839

**AOD trend over the Middle East**

K. Klingmüller et al.

groscopic growth (Abdelkader et al., 2015), increasing the particle size and thereby the AOD, similar to enhanced dust activity. These effects remain to be analysed in future studies. We plan a modelling study to further interpret the identified correlations in terms of causal relationships.

*Acknowledgements.* The research reported in this publication has received funding from King Abdullah University of Science and Technology (KAUST). For computer time, the resources of the KAUST Supercomputing Laboratory were used by the KAUST researchers. K. Klingmüller is supported by the KAUST CRG3 grant URF/1/2180-01 Combined Radiative and Air Quality Effects of Anthropogenic Air Pollution and Dust over the Arabian Peninsula.

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

## ACPD

doi:10.5194/acp-2015-839

**AOD trend over the Middle East**

K. Klingmüller et al.

**Table 1.** Small-sample-size corrected Akaike information criterion (AICc) and $R^2$ values for linear AOD models using different predictors. For each country, lower AICc values indicate preferable models. For Iran, precipitation is a more relevant predictor than soil moisture, suggesting greater relevance for atmospheric aerosol transport than in Saudi Arabia and Iraq. Only for Saudi Arabia, the model can be improved by including surface wind as predictor (last row). Corresponding scatter plots are shown in Fig. S30 in the Supplement.

| Predictors | Saudi Arabia | | Iraq | | Iran | |
|---|---|---|---|---|---|---|
| | AICc | $R^2$ | AICc | $R^2$ | AICc | $R^2$ |
| soil moisture | −54.0 | 0.82 | −49.4 | 0.89 | −82.4 | 0.84 |
| precipitation | −45.4 | 0.66 | −34.4 | 0.67 | −88.7 | 0.90 |
| soil moisture + precipitation | −52.0 | 0.85 | −52.1 | 0.94 | −90.5 | 0.94 |
| soil moisture + precipitation + wind | −55.2 | 0.93 | −47.6 | 0.94 | −86.0 | 0.94 |

**ACPD**

doi:10.5194/acp-2015-839

**AOD trend over the Middle East**

K. Klingmüller et al.

**Table 2.** Regression coefficients for the linear AOD model and their standard errors.

| | $\beta_0$ | $\beta_\theta$ | $\beta_P/(\mathrm{days\,mm^{-1}})$ |
|---|---|---|---|
| Saudi Arabia | $1.19 \pm 0.22$ | $-8.3 \pm 2.3$ | $-0.15 \pm 0.11$ |
| Iraq | $1.40 \pm 0.12$ | $-5.25 \pm 0.78$ | $-0.130 \pm 0.048$ |
| Iran | $0.346 \pm 0.059$ | $-1.07 \pm 0.44$ | $-0.060 \pm 0.015$ |

**ACPD**

doi:10.5194/acp-2015-839

**AOD trend over the Middle East**

K. Klingmüller et al.



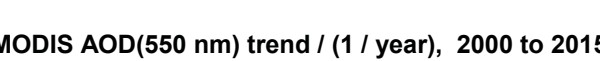

**MODIS AOD(550 nm) trend / (1 / year), 2000 to 2015**

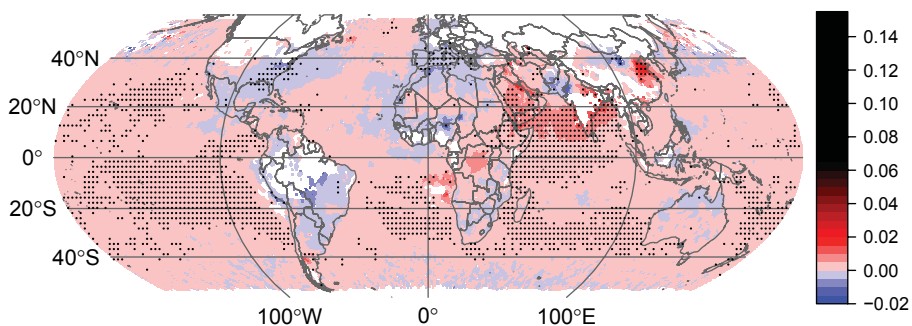

**Figure 1.** Global AOD trends between March 2000 and February 2015, based on the Dark Target/Deep Blue 550 nm AOD from MODIS Terra, collection 6. Regions with significant trends ($p$ value $< 0.01$) are dotted, regions with incomplete time series are coloured white. Despite being less pronounced than for the period January 2001 to December 2010 considered in (Pozzer et al., 2015) (see also Fig. S1 in the Supplement), the Middle East is a hot spot of AOD increase. Comparable trends are only found in China and, in a smaller area, near the Aral Sea. The exceptional trend in the latter region defines the upper limit of the colour scale.

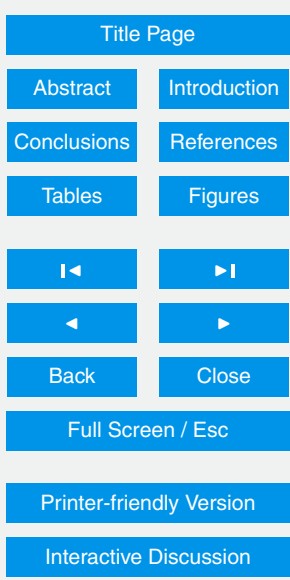

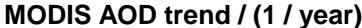

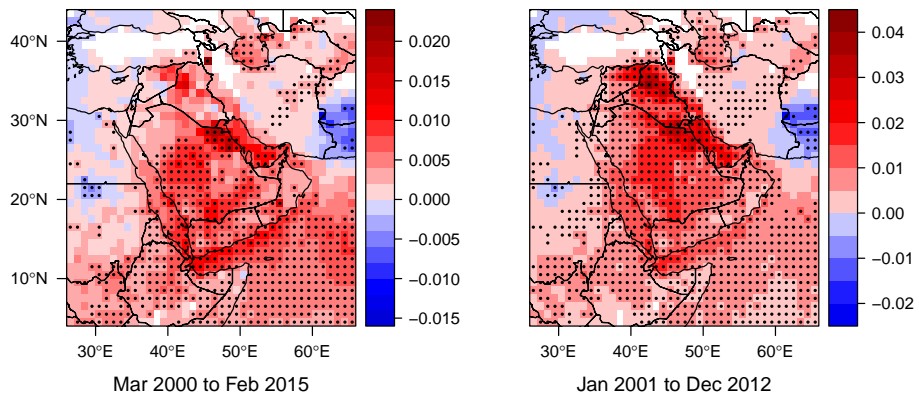

**Figure 2.** Pattern of Middle East AOD trends between March 2000 and February 2015 (left) and between January 2001 and December 2012 (right), based on the Dark Target/Deep Blue 550 nm AOD from MODIS Terra, collection 6. Pixels with significant trend ($p$ value $< 0.01$) are marked with a dot. The same plot for the period January 2001 to December 2010 is shown in Fig. S2 in the Supplement.

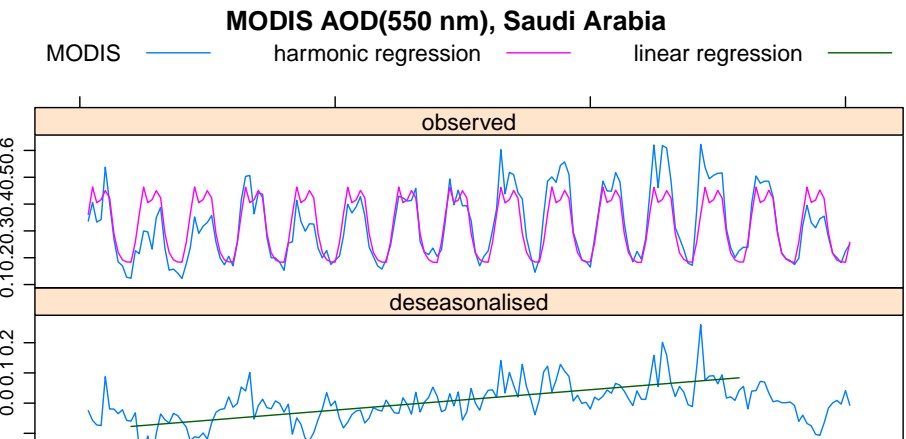

**Figure 3.** Evolution of the average 550 nm AOD over Saudi Arabia from March 2000 to February 2015. Monthly values from MODIS Terra Dark Target/Deep Blue, collection 6, are shown. A seasonal cycle (pink) is obtained by harmonic regression. Subtracting the seasonal cycle yields the deseasonalised AOD (lower panel), the trend of which is quantified using linear regression taking the one-month-lag autocorrelation of the time series into account (green).

Discussion Paper | Discussion Paper | Discussion Paper | Discussion Paper |

**ACPD**

doi:10.5194/acp-2015-839

**AOD trend over the Middle East**

K. Klingmüller et al.

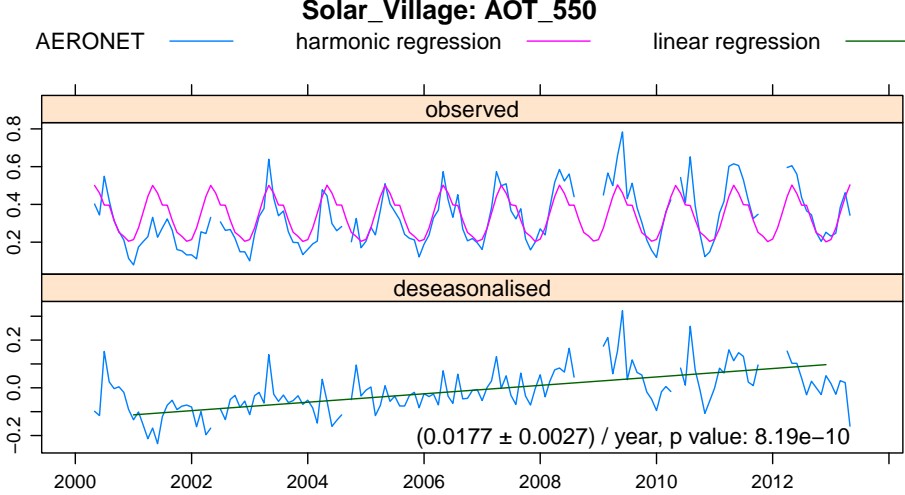

**Figure 4.** Evolution of the AOD measured by the AERONET station Solar Village. The measurements have been linearly interpolated to 550 nm wavelength. Deseasonalisation and trend analysis are performed as for Fig. 3.

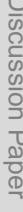

## ACPD

doi:10.5194/acp-2015-839

**AOD trend over the Middle East**

K. Klingmüller et al.

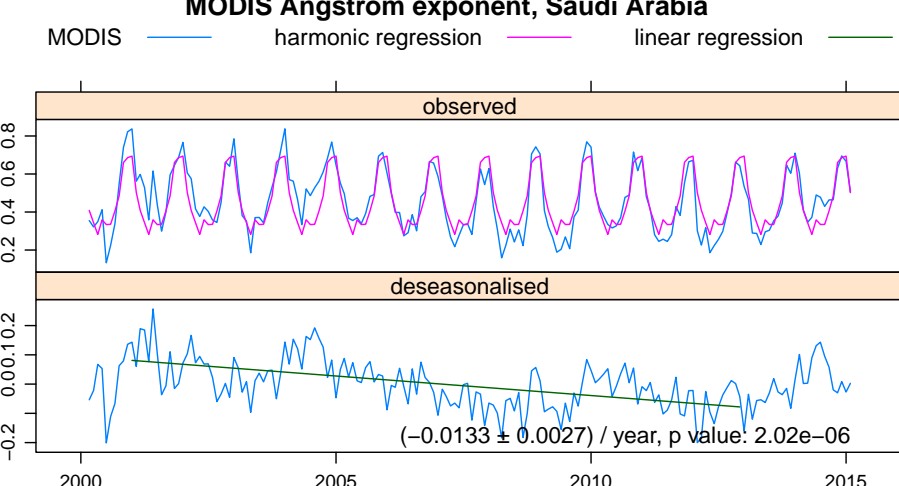

**Figure 5.** Evolution of the average Saudi Arabian Ångström exponent from March 2000 to February 2015. Monthly values from MODIS Terra Deep Blue, collection 6, are shown. Deseasonalisation and trend analysis are performed as for Fig. 3.

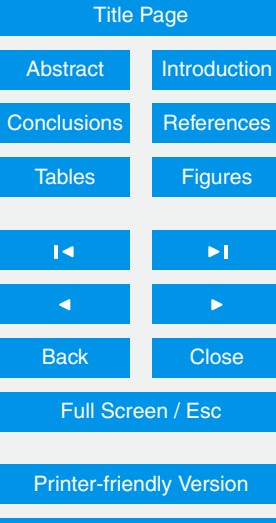

**ACPD**

doi:10.5194/acp-2015-839

**AOD trend over the Middle East**

K. Klingmüller et al.

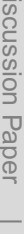

**ACPD**

doi:10.5194/acp-2015-839

**AOD trend over the Middle East**

K. Klingmüller et al.

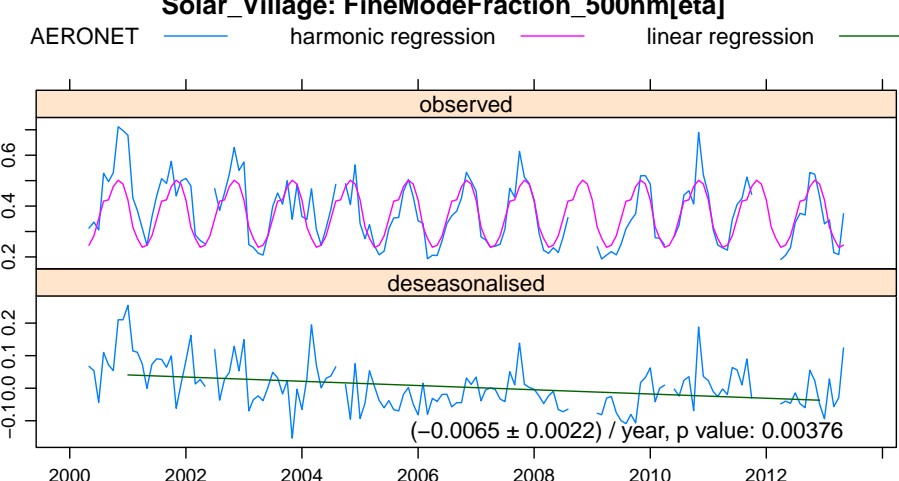

**Figure 6.** AERONET fine mode fraction at the station Solar Village. Deseasonalisation and trend analysis is performed as for Fig. 3.

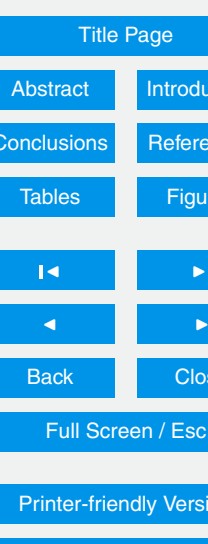

**ACPD**

doi:10.5194/acp-2015-839

**AOD trend over the Middle East**

K. Klingmüller et al.

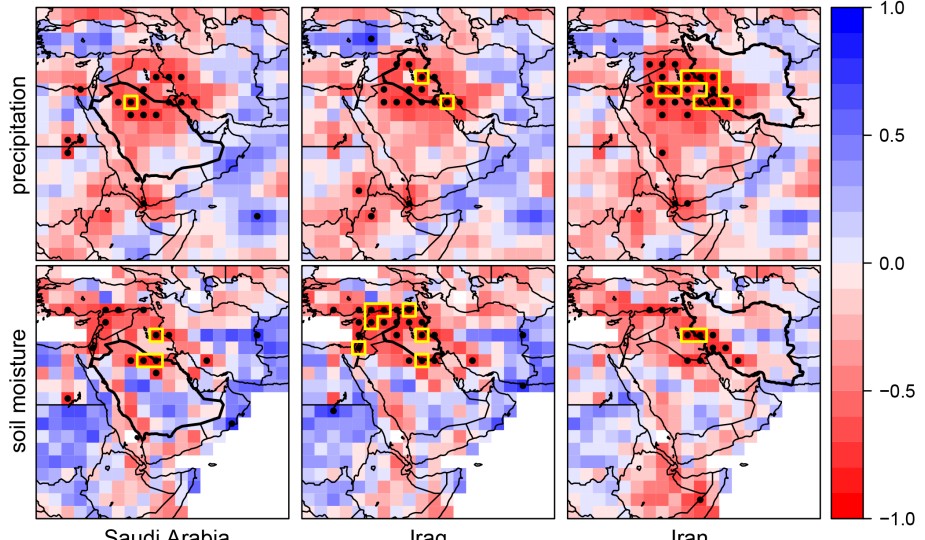

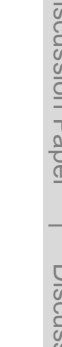

**Figure 7.** Correlation of precipitation (top row) and soil moisture (bottom row) with the AOD over Saudi Arabia, Iraq and Iran. For soil moisture and AOD, annual averages are used where the averaging period starts with the AOD season on 1 December, for the precipitation the averaging period starts with the precipitation season on 1 September to take effects on vegetation and soils into account. Pixels with significant correlation ($p$ value $< 0.01$) are marked with a dot. Pixels with correlation coefficients below $-0.8$ are outlined in yellow. MODIS Terra collection 6 combined Dark Target/Deep Blue 550 nm AOD data is used, precipitation data is taken from TRMM 3B42, soil moisture from the European Space Agency Climate Change Initiative (ESA-CCI).

Discussion Paper | Discussion Paper | Discussion Paper | Discussion Paper |

**ACPD**

doi:10.5194/acp-2015-839

**AOD trend over the Middle East**

K. Klingmüller et al.

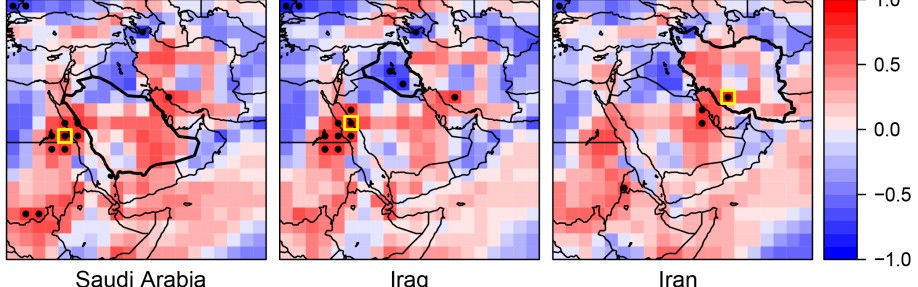

Saudi Arabia        Iraq        Iran



**Figure 8.** Correlation of surface wind with AOD over Saudi Arabia, Iraq and Iran. For both, wind and AOD, annual averages are used where the averaging period starts with the enhanced AOD season on 1 December. Pixels with significant correlation ($p$ value $< 0.01$) are marked with a dot. Pixels with correlation coefficients above 0.8 are outlined in yellow. MODIS Terra collection 6 combined Dark Target/Deep Blue 550 nm AOD data is used, for the surface wind we use the ERA-Interim wind at 10 m altitude.

**ACPD**

doi:10.5194/acp-2015-839

**AOD trend over the Middle East**

K. Klingmüller et al.

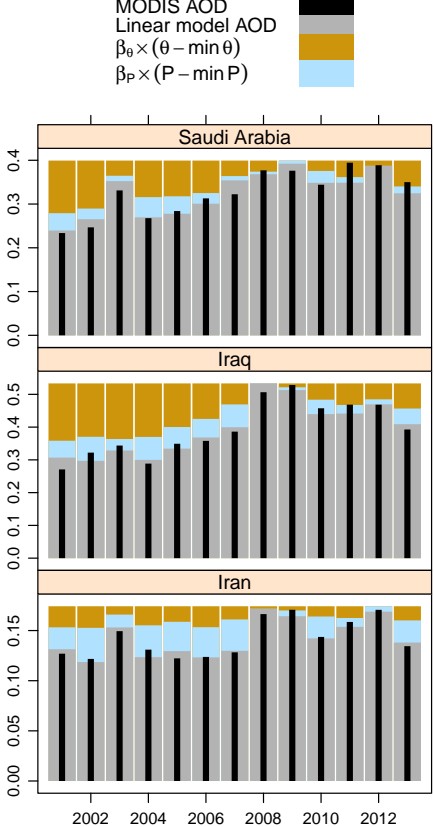

**Figure 9.** Comparison of MODIS observations (black) and linearly modelled AOD (grey). The brown (blue) bars represent the contribution of the soil moisture (precipitation) term to the interannual variability. For Saudi Arabia and Iraq, the soil moisture term is most strongly related to the AOD increase during the last decade, whereas the precipitation term adds variability. For Iran, the precipitation term is much more relevant, indicating that aerosol transport and associated precipitation scavenging dominates the aerosol variability.