# Peer review of "MODIS AOD(550 nm) trend / (1 / year), 2001 to 2010"

_Atmospheric Chemistry and Physics, 2015_

## Short Comment (SC1) · 15 Jan 2016

This is a short comment, not a full review of this paper. I am quite interested in the topic of the study and it is good to see the MODIS Collection 6 data products used in this way.

Although I know the main topic of the paper is AOD trends in the Middle East, there is some discussion, and a map (Figure 1), of trends globally. In this Figure the authors note very large positive trends (a factor of 3 or so larger than trends elsewhere) around the edge of the Aral Sea. If this trend is real, then it would certainly be an important result. Because it is such a strong result, I think it warrants some further examination, even though not the main focus of the study. I have a worry that it may be spurious and the casual reader of the paper may see it and read more in to it than is warranted.

The Aral Sea (and indeed much of that part of central Asia) is quite a difficult region

for space-borne AOD retrievals. The terrain can be quite complicated spatially and temporally, and there is an absence of validation data sources (e.g. AERONET) over much of the region, which means that the performance of the retrievals in this area is really not well-known (and has not to my knowledge been characterised directly before).

For the Aral Sea and surroundings in particular, there is a lot of seasonal and interannual variability in the surface cover: the shorelines have changed a lot over the past decades (even over the past 15 years), there can be temporary flooding/vegetation growth, and dry salty lake beds can get very bright (strong BRDF hotspot effect). In short, characterisation of the surface reflectance in this region is quite complicated, and errors in this can lead to artefacts in retrieved AOD, often over very localised areas and for short periods of time (e.g. a dry lake bed, or surface covered by a very thin layer of water, which happens to be viewed at a geometry close to that of specular reflection). These artefacts will most likely be apparent values of high AOD (as often they mean the surface is brighter than it is assumed to be). As another complication, sharp boundaries in surface cover in low-AOD conditions can sometimes lead to false positive cloud identification (i.e. data being thrown away as cloudy when it is really not cloudy) because some cloud mask tests are based on spatial variability of observed top-of-atmosphere reflectance. This poses risks to trend analyses because, in this area, there may be step changes or trends in surface cover which are not being accounted for well and manifesting as apparent trends in AOD.

A further complication arises from the MODIS Level 3 gridding strategy. As the Level 3 data are a simple mean of Level 2 data (within a single day, then averaged), a small number of very high-biased retrievals can throw off the daily or monthly average for a grid cell, particularly if the total number of retrievals is low.

I therefore advise caution in presentation of trends around the Aral Sea. I would suggest that the authors take a look at some of the time series for the individual grid cells where they see these trends, to check whether the trends are plausible. For example a

gradual consistent increase in AOD would be physically reasonable, while if the trend is coming from a single year or few years with AOD much higher than all earlier ones, it may be more likely the result of something like a change in surface type leading to a change in the error characteristics of the retrievals. The types of statistical tests used in these trend analyses may not always be able to identify when situations like this occur. Manual inspection of the time series may reveal individual months which stick out, at which point one can go back to daily data (either Level 2 or daily Level 3, in combination with true-colour images) to examine exactly what is going on and whether the retrievals seem realistic.

This is of course potentially an issue everywhere in the world, but our experience with the data suggests that regions of central Asia are among those where it is known to be a concern. One other thing the authors could consider doing here is to apply some threshold on the data volume within a month, to exclude poorly-sampled grid cells, since the MODIS monthly mean AOD product in such cases may not be a good representation of the true monthly mean. It would likely be a bit of trial and error to balance completeness of spatiotemporal sampling with the data volume remaining available for analysis, and I don't know whether it would help in this particular region, but it seems to me like it would be worth exploring.

The authors can feel free to contact me (Andrew Sayer, andrew.sayer@nasa.gov) if they have questions about the MODIS aerosol products.

---

## Short Comment (SC2) · 5 Feb 2016

I would like to thank the authors (Klingmüller et al) for the offline discussions which we have had in response to my initial comment: I am very appreciative of the extra efforts they have taken to dig into the data, and together I feel we've been able to identify more conclusively the source of the artefact over the Aral Sea. Hopefully we will be able to ameliorate this in future versions of our satellite aerosol data products.

As noted the Aral Sea was not the main focus of the authors' study, so hopefully the Discussion here will serve to explain this anomaly in the data but not distract readers from the main discussions of AOD trends elsewhere.

I wanted to take the opportunity to share some more of the offline discussions I have had with the authors here, for the interest of the journal readership.

Although I had initially suspected that the high-AOD artefacts in the data would mostly

be coming from the side of the over-land retrievals, it turns out that the artefacts are mostly coming from retrievals over nominal water surfaces. Some pixels identified as water by the MODIS land mask are ephemeral (sometimes water-covered, sometimes not) and not really suitable for AOD retrieval from an algorithm assuming a dark water surface. While some of these are caught and eliminated by the algorithm's internal quality checks, a few are not, and lead to high-biased AOD retrievals. Additionally, pixels identified as water but which are not actually covered by water for a given observation are skipped by the over-land AOD algorithms. So the situation here is that there are some areas which should not have retrievals but do have them, and some areas which should have retrievals but do not. Thus, in the Level 3 aggregated data, these artefacts become magnified.

The underlying cause is that the MODIS data have a static land mask, which is used in the aerosol products to identify which algorithm processing path (i.e. land or water) should be used for a specific pixel. In areas where the surface cover changes by more than a pixel over the MODIS mission (the Aral Sea being the most prominent, but not only, example), this can lead to situations where the use of the land mask in this way breaks down. Work is underway here to better identify regions of variable land/water cover like this in future versions of the MODIS data products. In the meantime the authors' suggestion to look at histogram statistics to identify potentially anomalous grid cells seems to work well for Level 3 data.

I have attached images from an example MODIS granule from last week which illustrates this (it is the most recent day in which I saw the problem). Panel (a) is a true-colour image. You can see that the western end of the South Aral Sea basin (which still contains water) is dark, but the part nearer the centre (the remaining eastern lobe) is very shallow and definitely does not fit a 'dark water' assumption. Panel (b) shows which algorithm contributes in each pixel to the 'combined' MODIS AOD data set. Over land here it is always Deep Blue; over water it is always the water algorithm (since there is only one).

Panels (c)-(e) show the Deep Blue, Dark Target land/ocean, and Combined AOD fields for this day. You can see the artefacts in the central/eastern over-water retrievals (the AOD exceeds 4 in a few places, in fact) compared to the more reasonable values elsewhere. So, when these data are gridded to the level 3 1 degree resolution, these artefacts will strongly dominate the grid cells. This is because some of the fully dried out parts of the Aral Sea are still counted as 'water' by the MODIS land mask, so no land retrievals are performed there.

It is over these ephemeral (either very shallow water or at this point dry – hard to tell from the image) water surfaces where sometimes it is doing an ocean retrieval when it should not. As a result the retrievals in the Level 3 grid cells over here are dominated by retrievals which should not have taken place. Which is also why the authors' histogram tests work.

I've addressed this issue to the extent currently possible in our forthcoming VIIRS Deep Blue land and ocean AOD data set (so it should become much less of a problem) and it is being worked on here within the MODIS team as well. Interested parties can feel free to contact me for more information.
* * *
(a) MODIS Aqua: 09:10 UTC, 27 Jan 2016

(b) Merged algorithm flag

Merged algorithm flag

Ocean   DB   Merged   DT

550 nm AOD

0   0.4   0.8   1.2   1.6   >2

(c) Deep Blue AOD, good QA

(d) Dark Target/ocean AOD, good QA

(e) Merged AOD, good QA

**Fig. 1.**

---

## Author Comment (AC1) · 5 Feb 2016

We thank Andrew Sayer for his most valuable comment on the MODIS AOD trend over the Aral Sea. Even though this region is not the main focus of this study, we agree that the corresponding results should not be mentioned without discussing the reliability of the retrievals over this specific terrain and scrutinising the resulting trend. We therefore propose the following additions to article and supplement:

PAGE 5, LINE 20

Amending

"Comparably strong trends are only found in China and, though restricted to a small area, under the exceptional conditions in the vicinity of the Aral Sea."

to

"Comparably strong trends are only found in China. Our analysis identifies even stronger trends over the Aral Sea region. However, the shrinking Aral Sea not only exposes new dust sources (Wiggs et al. 2003) but also implies a constantly changing surface reflectance making a consistent AOD retrieval over this region extremely challenging and likely contributing a large spurious component to the MODIS trend (Sayer 2016, see also the discussion in the supplement)."

CAPTION FIGURE 1 (PAGE 22)

Amending

"Comparable trends are only found in China and, in a smaller area, near the Aral Sea. The exceptional trend in the latter region defines the upper limit of the colour scale."

to

"Comparable trends are only found in China. An exceptional trend over the Aral Sea region defines the upper limit of the colour scale, but is likely predominantly attributed to retrieval artefacts (see the discussion in the supplement)."

SUPPLEMENT

We propose to add the pages in the supplement of this comment to the supplement of the article.

BIBLIOGRAPHY

New items:

Wiggs, G. F. S., O'hara, S. L., Wegerdt, J., Van Der Meer, J., Small, I. And Hubbard, R. (2003), The dynamics and characteristics of aeolian dust in dryland Central Asia: possible impacts on human exposure and respiratory health in the Aral Sea basin. The Geographical Journal, 169: 142–157. doi: 10.1111/1475-4959.04976

Sayer, A. M., personal communication, 2016, see also online discussion of Klingmueller et al. 2016

Please also note the supplement to this comment:
http://www.atmos-chem-phys-discuss.net/acp-2015-839/acp-2015-839-AC1-supplement.pdf
* * *
[Figure]

**Supplement:**

**Aral Sea AOD trends**

Our analysis identifies the strongest trend in the monthly level 3 MODIS AOD product (MODIS MOD08 M3) over the Aral Sea region. The dust activity in this region is increasing due to the drying Aral Sea which exposes new dust sources (Wiggs et al., 2003). However, due to rapidly changing shorelines and land cover, and the lack of ground based observations for validation, space-borne AOD retrievals over this region are very challenging (Sayer, 2016). This suggests that the extreme MODIS trend is not real in its full magnitude, but includes a spurious component due to retrieval artefacts, and motivates a more detailed analysis of the AOD data in question.

Figure Fig. S31 shows the time series of the one degree cell with the centre 44.5°N59.5°W in the Aral Sea region, which corresponds to the maximum in Fig. 1. The data exhibits a very constant growth over the 15 year period, i.e., the trend does not result from a few years of high AOD, and the small p value indicates a very high significance of the trend. What often characterises a reliable result for an AOD trend, in view of the shrinking Aral Sea rather appears to be a spurious drift related to the changing surface reflectance, in particular due to the changing shorelines. This is further supported by the extremely high AOD values towards the end of the period which suggest that the surface in reality is brighter than assumed for the retrieval algorithm.

Figure S32 shows that the high AOD values and the strong positive trend is restricted to only a few grid cells. The same cells suffer from a low number of AOD retrievals per month (Fig. S33), revealing difficult retrieval conditions. Moreover, in their AOD histograms, Fig. S34, the bin for large AOD values above 1.5 has a frequency comparable to that of bins for lower AOD values which under normal conditions does not occur. Based on this observation, a simple filter can be used to filter out the affected grid cells in the Aral Sea region with little impact on the retrievals globally: for considering a grid cell the frequency of the bin for large AODs (1.5 to 5) is required to be smaller than the mean frequency of all bins (0 to 0.1, 0.1 to 0.3, 0.3 to 0.6, 0.6 to 1.5, 1.5 to 5). As Fig. S35 shows, this condition is fulfilled almost globally. Applying the filter to Fig. 1 yields Fig. S36, where the AOD trend has an upper limit of about 0.025 per year.

**References**

Klingmüller, K., Pozzer, A., Metzger, S., Stenchikov, G., and Lelieveld, J.: Aerosol optical depth trend over the Middle East, Atmospheric Chemistry and Physics Discussions, 2016, 1–30, doi:10.5194/acp-2015-839, http://www.atmos-chem-phys-discuss.net/acp-2015-839/, 2016.

MODIS MOD08 M3: ftp://ladsweb.nascom.nasa.gov/allData/6/MOD08_M3/, visited 31 Aug 2015.

Sayer, A. M.: personal communication, see also online discussion of Klingmüller et al. (2016), 2016.

Wiggs, G. F. S., O'hara, S. L., Wegerdt, J., Van Der Meer, J., Small, I., and Hubbard, R.: The dynamics and characteristics of aeolian dust in dryland Central Asia: possible impacts on human exposure and respiratory health in the Aral Sea basin, Geographical Journal, 169, 142–157, doi:10.1111/1475-4959.04976, http://dx.doi.org/10.1111/1475-4959.04976, 2003.

[Figure]

**Figure S31.** Same as Fig. 3, but for the one degree cell with the centre 44.5°N59.5°W in the Aral Sea region.

[Figure]

**Figure S32.** Each panel displays the 15 year AOD time series for a one degree grid cell the location of which is indicated by the grey silhouette showing the Aral Sea. Figure S31 corresponds to the third pixel in the fourth row. Some cells have incomplete monthly time series and therefore are not shown in Fig. 1.

[Figure]

**Figure S33.** Same as Fig. S32 but showing the number of retrievals per month instead of the AOD.

[Figure]

**Figure S34.** 15 year averages of the monthly AOD histograms. Each panel shows the histogram for the one degree grid cell the location of which is indicated by the grey silhouette showing the Aral Sea.

[Figure]

**Figure S35.** In light yellow grid cells the frequency of high AOD values above 1.5 is smaller than the average frequency of the AOD bins 0 to 0.1, 0.1 to 0.3, 0.3 to 0.6, 0.6 to 1.5 and 1.5 to 5. In dark red cells this condition is not fulfilled, which raises the suspicion that the retrievals over these regions are unreliable and gives reason to not consider them for further analysis.

[Figure]

**Figure S36.** As Fig. 1, but after applying the filter shown in Fig. S35.

---

## Referee Comment (RC1) · Anonymous Referee #1 · 13 Feb 2016

The manuscript presents an analysis of trends in the aerosol optical depth and other variables over Iraq, Iran, and Saudi Arabia from 2000 to 2015, which can be mostly ascribed to changes in the dust load over the region in the time period. Using the tool of multiple linear regression, a first analysis of attribution of the trends to soil moisture, precipitation, and surface wind speed is provided too. The authors use multiple updated data sets and sound methodology. The whole paper is straightforward and very well written, and the conclusions are convincing. The results are new, relevant, and appropriate for the journal. I only have one point to make, which should be addressed, before publication:

On page 5, the authors estimate the AOD trend by fitting a linear model, taking into account the autocorrelation with a time lag of one months. How was it determined that this choice was sufficient to take autocorrelation into account? Did the authors test the autocorrelation structure of the residuals for different time lags, whether it approximates

the one of white noise? I recommend to add such a test to the manuscript.

---

## Referee Comment (RC2) · Anonymous Referee #2 · 22 Mar 2016

This manuscript may be accepted for publication.

---

## Author Comment (AC2) · 12 Apr 2016

We thank the referee for reviewing the manuscript, the positive comments and the valuable remark on the autocorrelation.

Regarding the time lag considered in the autoregressive model, for consistency with previous studies we follow the approach of de Meij et al. (2012), Hsu et al. (2012) and Pozzer et al. (2015), who employ the method described by Weatherhead et al. (1998) and the references therein (Tiao 1990), and use a first order autoregressive (AR(1)) model for monthly data.

We propose to add the supplement of this comment to the supplement of the article. For all time series plots in the manuscript (Figs. 3 to 6) it includes plots of the partial autocorrelation functions (Figs. S31, S33, S35, S37) showing that only the first order (1 month lag) term is significant. Additionally, Figs. S32, S34, S36 and S38 display

the correlograms of the residuals of the corresponding AR(1) models, revealing that the residuals are well approximated by white noise and providing further evidence that using the first order model sufficiently accounts for the autocorrelation structure.

In the main text we propose the following additions:

PAGE 5, LINE 15

Extending

"The trend of the deseasonalised AOD ist calculated by fitting a linear model using generalised least squares taking into account the time series auto correlation with a time lag of one month (Pinheiro et al., 2015)."

to

"The trend of the deseasonalised AOD has been calculated by fitting a linear model using generalised least squares (Pinheiro et al., 2015). Consistent with Weatherhead et al. (1998), de Meij et al. (2012), Hsu et al. (2012) and Pozzer et al. (2015), the one-month-lag autocorrelation was used to account for the autocorrelation structure of the time series, see Figs. S31 to S38 in the Supplement."

PAGE 19

New reference item

Weatherhead, E. C., et al. (1998), Factors affecting the detection of trends: Statistical considerations and applications to environmental data, J. Geophys. Res., 103(D14), 17149–17161, doi:10.1029/98JD00995.

PAGE 24, CAPTION FIGURE 3

Adding

"The autocorrelation structure of the time series is studied in Figs. S31 and S32 in the Supplement."

PAGE 25, CAPTION FIGURE 4

Adding "; see also Figs. S33 and S34 in the Supplement"

PAGE 26, CAPTION FIGURE 5

Adding "; see also Figs. S35 and S36 in the Supplement"

PAGE 27, CAPTION FIGURE 6

Adding "; see also Figs. S37 and S38 in the Supplement"

References:

A. de Meij, A. Pozzer, J. Lelieveld, Trend analysis in aerosol optical depths and pollutant emission estimates between 2000 and 2009, Atmospheric Environment, Volume 51, May 2012, Pages 75-85, ISSN 1352-2310, http://dx.doi.org/10.1016/j.atmosenv.2012.01.059.

Hsu, N. C., Gautam, R., Sayer, A. M., Bettenhausen, C., Li, C., Jeong, M. J., Tsay, S.-C., and Holben, B. N.: Global and regional trends of aerosol optical depth over land and ocean using SeaWiFS measurements from 1997 to 2010, Atmos. Chem. Phys., 12, 8037-8053, doi:10.5194/acp-12-8037-2012, 2012.

Pozzer, A., de Meij, A., Yoon, J., Tost, H., Georgoulias, A. K., and Astitha, M.: AOD trends during 2001–2010 from observations and model simulations, Atmos. Chem. Phys., 15, 5521-5535, doi:10.5194/acp-15-5521-2015, 2015.

Tiao, G. C., G. C. Reinsel, D. Xu, J. H. Pedrick, X. Zhu, A. J. Miller, J. J. DeLuisi, C. L. Mateer, and D. J. Wuebbles (1990), Effects of autocorrelation and temporal sampling schemes on estimates of trend and spatial correlation, J. Geophys. Res., 95(D12), 20507–20517, doi:10.1029/JD095iD12p20507.

Weatherhead, E. C., et al. (1998), Factors affecting the detection of trends: Statistical considerations and applications to environmental data, J. Geophys. Res., 103(D14),

17149–17161, doi:10.1029/98JD00995.

Please also note the supplement to this comment:
http://www.atmos-chem-phys-discuss.net/acp-2015-839/acp-2015-839-AC2-supplement.pdf

[Figure]

**Supplement:**

**Partial correlogram (MODIS AOD(550 nm), Saudi Arabia)**

[Figure]

**Figure S31.** Partial correlogram of the deseasonalised MODIS AOD time series over Saudi Arabia in Fig. 3. The dashed lines represent the confidence limits for a significance level of 5%. Only the 1 month lag partial autocorrelation is unambiguosly significant, suggesting that the time series follows an AR(1) process. The absence of a significant partial autocorrelation for a 12 month lag demonstrates the good performance of the deseasonalisation procedure.

**AR(1) residual correlogram (MODIS AOD(550 nm), Saudi Arabia)**

[Figure]

**Figure S32.** Correlogram for the residuals of the AR(1) model fitted to the deseasonalised MODIS AOD time series over Saudi Arabia in Fig. 3. The dashed lines represent the confidence limits for a significance level of 5%. No significant autocorrelations are found, providing evidence that the residuals are well approximated by white noise and that the AR(1) model yields a good approximation of the deseasonalised time series.

**Partial correlogram (AERONET AOD, Solar Village)**

[Figure]

**Figure S33.** As Fig. S31, but for the AERONET AOD over Solar Village shown in Fig. 4.

**AR(1) residual correlogram (AERONET AOD, Solar Village)**

[Figure]

**Figure S34.** As Fig. S32, but for the AERONET AOD over Solar Village shown in Fig. 4.

[Figure]

**Figure S35.** As Fig. S31, but for the MODIS Ångström exponent shown in Fig. 5.

[Figure]

**Figure S36.** As Fig. S32, but for the MODIS Ångström exponent shown in Fig. 5.

**Partial correlogram (AERONET fine mode fraction, Solar Village)**

[Figure]

**Figure S37.** As Fig. S31, but for the AERONET fine mode fraction over Solar Village shown in Fig. 6.

**AR(1) residual correlogram (AERONET fine mode fraction, Solar Village)**

[Figure]

**Figure S38.** As Fig. S32, but for the AERONET fine mode fraction over Solar Village shown in Fig. 6.

---

## Author Comment (AC3) · 12 Apr 2016

We thank the referee for reviewing the manuscript and the positive comment.

---

## Author Response (AR1)

**Relevant changes**

In accordance with the author comments AC1 and AC2, the following changes have been applied to manuscript and supplement. The changes to the manuscript are highlighted in the latexdiff output further below.

**PAGE 5, LINE 15**

Extended

"The trend of the deseasonalised AOD ist calculated by fitting a linear model using generalised least squares taking into account the time series auto correlation with a time lag of one month (Pinheiro et al., 2015)."

to

"The trend of the deseasonalised AOD has been calculated by fitting a linear model using generalised least squares (Pinheiro et al., 2015). Consistent with Weatherhead et al. (1998), de Meij et al. (2012), Hsu et al. (2012) and Pozzer et al. (2015), the one-month-lag autocorrelation was used to account for the autocorrelation structure of the time series, see Figs. S31 to S38 in the Supplement."

**PAGE 5, LINE 20**

Extended

"Comparably strong trends are only found in China and, though restricted to a small area, under the exceptional conditions in the vicinity of the Aral Sea."

to

"Comparably strong trends are only found in China. Our analysis identifies even stronger trends over the Aral Sea region. However, the shrinking Aral Sea not only exposes new dust sources (Wiggs et al. 2003) but also implies a constantly changing surface reflectance making a consistent AOD retrieval over this region extremely challenging and likely contributing a large spurious component to the MODIS trend (Sayer 2016, see also the discussion in the supplement)."

**PAGE 22, CAPTION FIGURE 1**

Extended

"Comparable trends are only found in China and, in a smaller area, near the Aral Sea. The exceptional trend in the latter region defines the upper limit of the colour scale."

to

"Comparable trends are only found in China. An exceptional trend over the Aral Sea region defines the upper limit of the colour scale, but is likely predominantly attributed to retrieval artefacts (see the discussion in the supplement)."

**PAGE 24, CAPTION FIGURE 3**
Added

"The autocorrelation structure of the time series is studied in Figs. S31 and S32 in the Supplement."

**PAGE 25, CAPTION FIGURE 4**
Added "; see also Figs. S33 and S34 in the Supplement"

**PAGE 26, CAPTION FIGURE 5**
Added "; see also Figs. S35 and S36 in the Supplement"

**PAGE 27, CAPTION FIGURE 6**
Added "; see also Figs. S37 and S38 in the Supplement"

[revised manuscript text omitted]